# Combining aerial photos and LiDAR data to detect canopy cover change in urban forests

**Kathleen Coupland**[1]*, **David Hamilton**[2], **Verena C. Griess**[1,3]

**1** Faculty of Forestry, Department of Forest Resources Management, University of British Columbia, Forest Sciences Centre, Vancouver, Canada, **2** College of Forestry, Oregon State University, Corvallis, OR, United States of America, **3** Department of Environmental System Sciences, Institute of Terrestrial Ecosystems, Eidgenössische Technische Hochschule Zürich, Universitätstrasse, Zürich, Switzerland

* kathleen.coupland@ubc.ca

**Data Availability Statement:** 2015 LiDAR data are available at the UBC Abacus data network (https://hdl.handle.net/11272.1/AB2/KET75X). 1949 UBC geography air photos are available in the collection archive (https://gic.geog.ubc.ca/resources/air-

## Abstract

The advancement and accessibility of high-resolution remotely sensed data has made it feasible to detect tree canopy cover (TCC) changes over small spatial scales. However, the short history of these high-resolution collection techniques presents challenges when assessing canopy changes over longer time scales (> 50 years). This research shows how using high-resolution LiDAR data in conjunction with historical aerial photos can overcome this limitation. We used the University of British Columbia's Point Grey campus in Vancouver, Canada, as a case study, using both historical aerial photographs from 1949 and 2015 LiDAR data. TCC was summed in 0.05 ha analysis polygons for both the LiDAR and aerial photo data, allowing for TCC comparison across the two different data types. Methods were validated using 2015 aerial photos, the means (Δ 0.24) and a TOST test indicated that the methods were statistically equivalent (±5.38% TCC). This research concludes the methods outlined is suitable for small scale TCC change detection over long time frames when inconsistent data types are available between the two time periods.

## Introduction

There is a growing body of research examining urban environments, including the benefits provided by urban forests and trees [1–4]. Urban forests offer many benefits to community members including, better quality of life [2,5], health benefits [4], psychological benefits, aesthetic benefits, and mitigation of pollutants [6]. They provide recreational space and allow people to experience nature within local settings [6].

Despite being of great value, urban forests are constantly under threat of development as densification and urban expansion continue [1]. Continuing urbanization has raised concerns about the loss of urban tree canopy cover (TCC) on global and regional scales [7,8]. A study looking at 2002 to 2009 across the United States showed that there was an average TCC decline of 0.2% per 6 years [8]. The results of this study have been further corroborated by a 2018 study that examined TCC changes across all of the United States and showed that 23 states had statistically significant declining TCC and 22 others showed a declining trend, although the decline was not statistically significant [9]. Monitoring urban TCC is important to making

photo-collection-and-services/). Photos to request: BC792:6, BC728:81, BC792:7, BC728:5, BC728:7, BC737:64, BC737:63, BC737:61, BC728:9. 2015 images were retrieved from Google Earth Pro 7.3. (May 19, 2015) Metro Vancouver Regional District, B.C, Canada. 49°15'38.21"N, 123°14'15.90"W (centroid coordinates). No labels, Historical Imagery. The authors confirm that they had no special privileges in accessing these datasets which other interested researchers would not have.

**Funding:** KC - Award # 6372 from the University of British Columbia. Award is funding for a PhD. The funders had no role in study design, data collection and analysis, decision to publish, or preparation of the manuscript.

**Competing interests:** The authors have declared that no competing interests exist.

informed management and development decisions to ensure that the benefits of urban forests are retained. Beyond identifying TCC changes, mapping these changes allows for spatially explicit visualization of urban development and its impacts on the urban forests [3], allowing for targeted decision making to ensure that development does not unduly impact the urban forest in already adversely impacted areas. This mapping can also aid in the identification of areas that are at risk of deforestation or in need of reforestation or afforestation, aiding future development and management decisions.

TCC is the percentage of an area covered by tree canopies and is the most common measurement for assessing urban forests, in part because it is easily understood by members of the public and it is a simple proxy to measure the amount of urban forest [3,7]. Urban forests are defined as "forest stands and trees with amenity values situated in or near urban areas" [10]. Various approaches and data sources have been used to estimate TCC, including field sampling, aerial photography interpretation, satellite imagery, Light Detection and Ranging (LiDAR) from both manned and unmanned machines [11]. Each of these methods have differences in costs, resolution, and time ranges. Remotely sensed data has increased in use for TCC measurements because it is often more cost effective than field surveys and sampling [12]. Change detection over time requires a minimum of two data sets from different periods of time regardless of the methodology used to determine and detect the changes [13].

Examination of time periods outside of new multispectral and point cloud data collection methods traditionally rely on access to historical aerial photos or manually collected field data [14–16]. The accessibility of aerial photos in many major urbanized areas and the ability to re-analyze the images using modern technology make it the predominate choice for historical TCC assessments [1,8]. Air photo interpretation has been used to determine urban TCC through using digitally orthorectified images due to its simplicity and availability of low-cost data [3]. The numerous advances in multispectral and point cloud data access, collection, and storage is occurring concurrently with increased computer processing improving canopy classification speed and fidelity [17,18]. This has precipitated contemporary remotely sensed TCC assessments to trend away from single spectrum aerial photo data towards these newer data types.

Landsat satellites despite providing the longest continuous series of remotely sensed global images beginning in 1972, is limited by its short legacy and is unable to examine changes over time scales that are available from other non-satellite sources. Although 50 years is a long time-scale in some industries (e.g. medicine, education, technology development) forestry often uses planning horizons spanning 100–300 years. Having data spanning the entirety of these planning horizons is often impossible and forest managers rely on data modeling to extrapolate forest changes and growth for decision making processes. Combining data sets and types would allow for TCC to be examined over longer time frames, reducing the need for extrapolation. For data continuity, many TCC change assessments use 1972 or 73 as the earliest year of study as this is when Landsat started delivering imagery [19–21]. Landsat is further restricted from small scale use because of the large (30x30m) pixel size. MODIS and RapidEye are other commonly used satellite scanners that are used for TCC and face similar issues as Landsat [22–24]. Starting in 1999, MODIS produces images with spatial resolutions ranging from 250 m to 1 km. RapidEye is a group of 5 satellites launched in 2008 and has a spectral resolution as fine as 5m. While RapidEye has finer a finer spatial scale than Landsat it is still limited by its recent initiation date. Additionally, RapidEye, much like other commercial satellites (including those with finer scale, e.g. Worldview, Planet, etc.) have financial costs associated with data access, limiting accessibility.

Unlike satellite scanners (Landsat, MODIS, RapidEye), laser scanners have advanced to allow for spatial resolutions of 10 cm (or less), allowing for examinations of individual tree

crowns, shapes and forms [25]. LiDAR, like all active sensors produce images without influence of shadows or light variations, which are common error sources in satellite and aerial imagery [26]. LiDAR's resolution abilities and lack of light interference make it optimal for the examination of tree canopies in urban areas where there are an intimate mixture of trees and human development. However, LiDAR has only recently been able to achieve such high resolutions and as adoption has increased the cost of collection has decreased making it more feasible.

Presently, in the assessment of TCC detection there is a dichotomy occurring between research examining small-scale spatial changes over short time periods (< 50 years) [8,9], and large-scale spatial change examining timescales within 1973 –present [27–29]. However, there are limited studies that have tried to combine canopy data types to allow for longer period (> 50 years) examinations of TCC on small scales. This study aims to determine the validity of combining historical aerial photos with LiDAR data as a method for detecting small-scale changes in urban TCC over time frames outside of LiDAR history.

## Data and methods

### Study area

The University of British Columbia's Vancouver Campus (UBC) (Canada) was selected as the study site based on data availability. A comparably large collection of historical aerial photos dating back to 1949 was available, as well as high resolution LiDAR data obtained in 2015. The study area is situated at the Western most tip of Vancouver's Point Grey Peninsula. The historically dominant tree species are *Tsuga heterophylla*, *Thuja plicata*, and *Pseudotsuga menziesii*. While the historical dominant species are still present in large quantities, there are also many non-native and planted species that occupy part of the urban forest. In addition to changes in the urban forest UBC has seen increases in enrollment and associated development to handle the growing population. In the last 60 to 70 years, student enrollment has grown almost tenfold from 7,960 to over 61,000. This increase in population has driven a demand for additional housing, leading to accelerated urbanization and drastic changes in TCC. The study area is constrained by the flight area of the LiDAR survey, which covered a total of 855.1 ha (Fig 1).

### Data collection and processing

Aerial Photos– 1949 and 2015. Historical aerial photos of the study area were acquired from the Air Photo collection at UBC's Geographic Information Centre. Photos were scanned manually at a resolution of 1200x1200 dpi and imported into ArcMap 10.6.1. Ground resolution of the photos was calculated at 11.64cm which is sufficient to examine TCC. All aerial photos were from April 1949. A spring month was selected because leaf-on seasonal photos are easier for aerial photo interpretation of TCC [3]. White borders were manually cropped off the images. Photos were georeferenced and rectified in ArcMap using ground control points and 2019 Google Earth imagery. A minimum of 10 ground control points were used, and a maximum root mean square error of 2m per image. A total of 10 georeferenced images were stitched together to create one image using ArcMap that covered the entire study area. In cases where images overlapped the image closest to vertical were used. 2015 georeferenced aerial photos were exported from Google Earth's historical imagery and imported into ArcMap. The 2015 images were resampled to have the same cell size as the 1949 images decreasing the resolution by about one third. They were stitched together and cropped to the study area.

**LiDAR.** The LiDAR data for this study was collected on May 20th 2015 spanning the study area was used for this analysis (University of British Columbia. Campus and Community Planning 2015). The dataset has a point spacing of 0.143m and a point density of 49.05 pts/m$^2$,

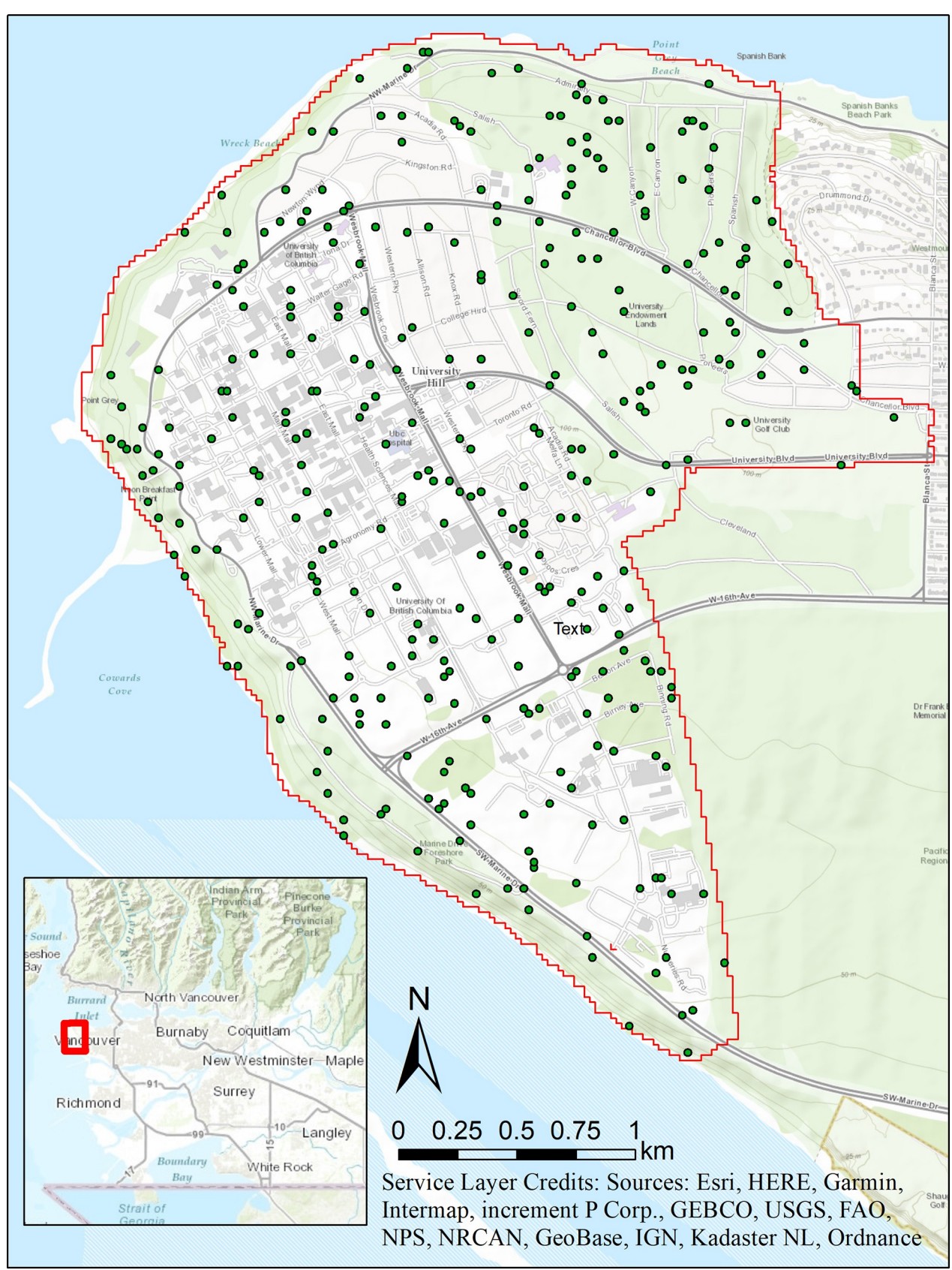

**Fig 1. Map of UBC showing the study extent and location of 342 randomly selected polygons used in the validation process.** Base map layer credit: Esri, HERE, Garmin, Intermap, increment P Corp., Gebco, USGS, FAO, NPS, NRCAN, GeoBase, ING (ArcGIS Licence 10.8.1).

exceeding the minimum resolution requirements for single tree canopy assessment [30]. The data was pre-processed; buildings were removed, and a canopy height model was created. Buildings were removed using ESRI's "Classify LAS Buildings". The canopy height model used a minimum height of 5m to remove shrubs and small plants [31] and had a 1m resolution.

## Methods

TCC for aerial photos was determined using the Tree Cover Mapping tool and methods from the USGS tree cover mapping tool manual [32]. The Tree Cover Mapping tool allows users to map canopy cover using visual interpretation of high-resolution photo imagery. Using a systematic grid, the user can estimate the TCC of each sample unit. A sample unit size of 0.05 ha was selected. The sample unit size was selected to stay consistent with previous research by Coupland et al. [34]. A calibration overlay of 10x10 dots was used to ensure accuracy and consistency of TCC values. Dots that intersected with tree canopies were summed, with each dot representing 1% canopy cover, to determine the canopy cover for each sample unit. Shape, size, texture and association was used to reduce the inclusion of shrubs or trees less than 5 m. Aerial photo interpretation was done by an single interpreter, with input available from other researchers familiar with aerial photos if object association was doubted. A TCC percentage was assigned to each of the 17 113 analysis polygons of 0.05 ha area.

LiDAR TCC was calculated using the R-package "Forest Tools" [33], and followed the forest cover classification methods from Coupland et al.'s research connecting urban TCC with forestry learning objectives [34]. Forest tools delineated tree canopies from CHM produced from high-density point clouds by identifying local maxima and creating a window radius to look for the surrounding local minimum. The radius size is dependent upon the height of the local maximum and canopy edges are delineated using the minima surrounding each maximum. As such, individual canopies are outlined and canopy polygons are created. The same 0.05 ha grid used in the aerial photo analysis was overlaid onto the LiDAR derived canopy polygons. The canopy polygons for each 0.05 ha section of the grid was summed and converted to a TCC percent.

## Validation

To ensure validity of comparing the 1949 aerial photos with LiDAR data, 2015 aerial photos were compared with the 2015 LiDAR data and examined for equivalency. We randomly selected 2% (342 polygons) of the 0.05 ha polygons randomly for validation (Fig 1). TCC for the 2015 aerial photos was determined using the same methods as for the analysis of 1949 aerial photos. Descriptive statistics, including a comparison of the mean and quantile-quantile (Q-Q) plot were completed. A Shapiro-Wilk test, a Wilcoxon-Signed rank test and a two-one-sided $t$-tests (TOST) equivalency test was performed to determine the validity of comparing the TCC between the two data sources. All statistical tests assumed $\alpha = 0.1$ and $\beta = 0.8$.

## Results

### Validation results

The mean TCC percent for the validation polygons were 54.94 for the 2015 aerial photos and 54.70 for the 2015 LiDAR data (Fig 2). The Q-Q plots were visually examined for normality (Fig 3). The Q-Q plots showed heavy-tails indicating a non-normal distribution with values at

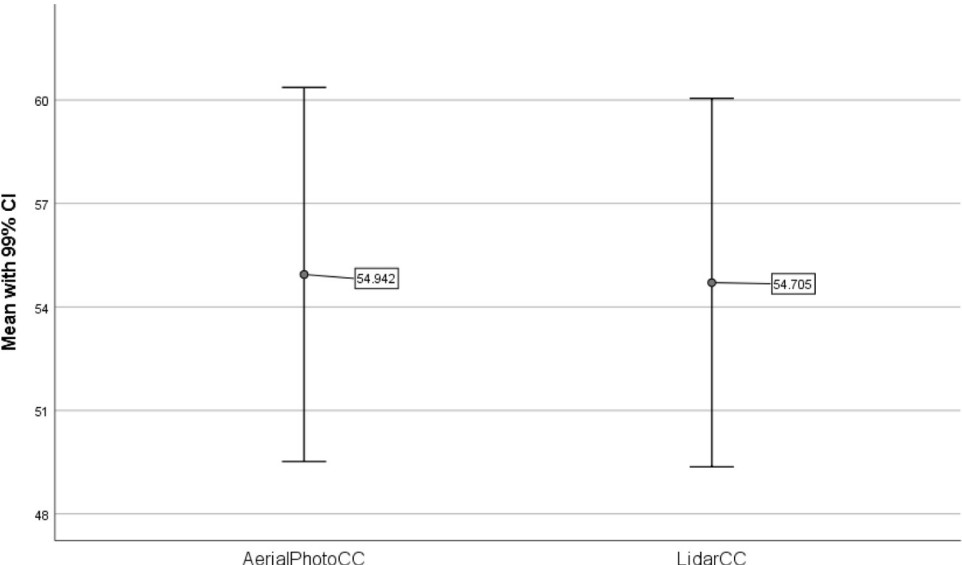

**Fig 2. Comparison of the means with 99% confidence interval between 2015 canopy cover using the USGS tree cover mapping tool and methods (AerialPhotoCC) and LiDAR analysis (LidarCC).**

both extremes (<10% and >90%) being more common. However, both Q-Q plots followed identical patterns indicating that the results from both methods picked up the same data trend of heavy-tails.

The Shapiro-Wilk test showed that the data for both the 2015 aerial photos and the 2015 LiDAR analysis were not normally distributed ($p < 0.000$), corroborating the visual analysis of the Q-Q plots. The non-parametric Wilcoxon-Signed rank test was used to test if the means canopy cover values from the 2015 aerial photo's canopy cover and the LiDAR data we different. We found that $p = 0.238$, suggesting the means were not significantly different. However, lack of statistical differences does not equate to statistical similarity.

To further test for equivalency a TOST equivalence test using $H_{01}$: $-\Delta < \mu_1 - \mu_2 > \Delta$. Where $\Delta$ is 6% TCC and $\mu_1 - \mu_2$ is the difference between the 2015 areal photo TCC and the 2015 LiDAR TCC. The TOST test based on Welch's t-test showed that the overserved effect size ($d = 0.01$) was significantly within the equivalent bounds of $d_{cohen} = \pm 0.14$, $t(681.85) = -1.75$ and $p = 0.04$. These results indicate that the methods are similar are within a range $\pm 5.38\%$ TCC and were deemed equivalent. We concluded that the methods used produced statistically equivalent results, allowing a comparison between the two data sources validating the methods and consequently allowing for comparison of TCC between the 1949 aerial photos and the 2015 LiDAR data.

## Canopy cover change results

TCC analysis of the 1949 aerial photos showed a mean TCC of 58.8% and the 2015 LiDAR showed a mean TCC of 49.9%. Since the 2015 aerial photo analysis and the 2015 LiDAR analysis were statistically equivalent, the differences between the 1949 aerial photo analysis and the 2015 LiDAR analysis represent realized changes in the TCC between the two time periods. A differences map (Fig 4) was created, with red and green showing deforestation and reforestation respectively, yellow indicated no change. These differences can be divided into three general trend areas: developed pre-1949, modern development, and coastal. The location of these three trend areas are roughly indicated in Fig 4.

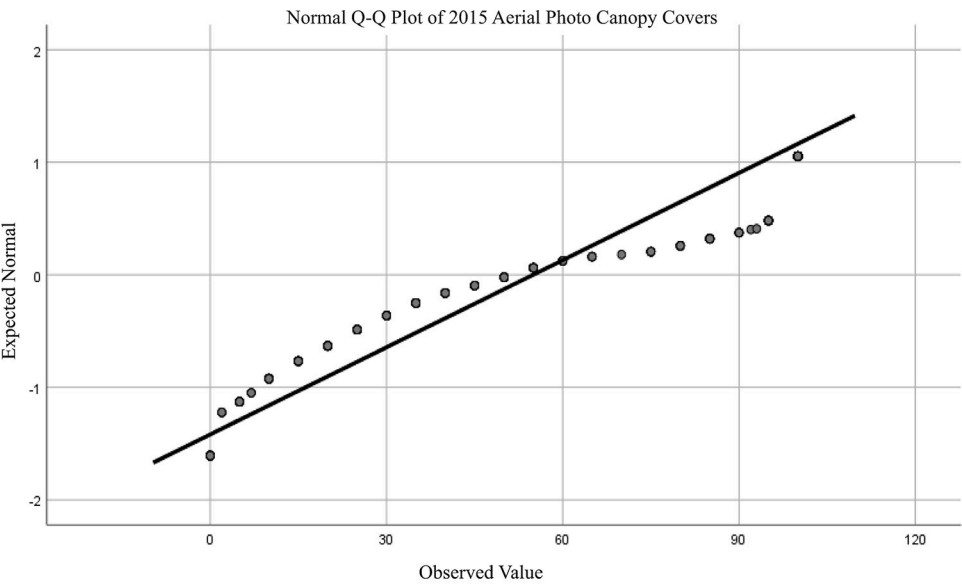

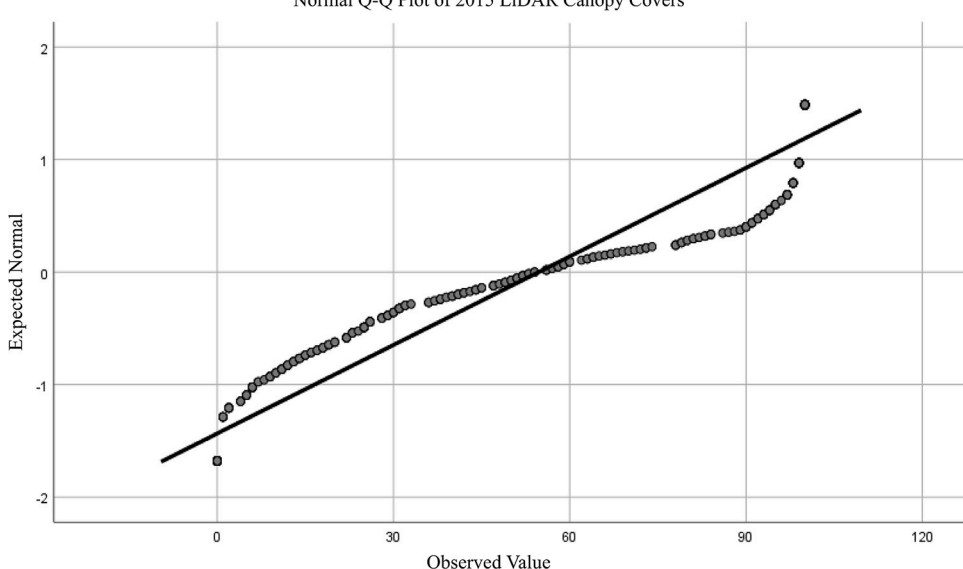

**Fig 3. Quantile-quantile normality plots for the 2015 aerial photo canopy covers and the 2015 LiDAR analysis.**
The plots are "heavy-tailed" indicating that the data is not normally distributed.

## Discussion

TCC changes were successfully detected for the 64-year period through combining 1949 aerial photo and 2015 LiDAR data derived canopy covers. Two different data types were used and we demonstrated that urban TCC can be compared using the methods presented despite differences in data types. This allowed examining TCC over a longer time period. Being able to compare TCC using two different data types potentially expands the timeframes available for other studies examining changes in urban forests over time.

TCC in areas that would naturally be forested have tree cover ranges from 45–65%, which is consistent with the mean values from both 1949 and 2015 [35]. The non-normal distribution

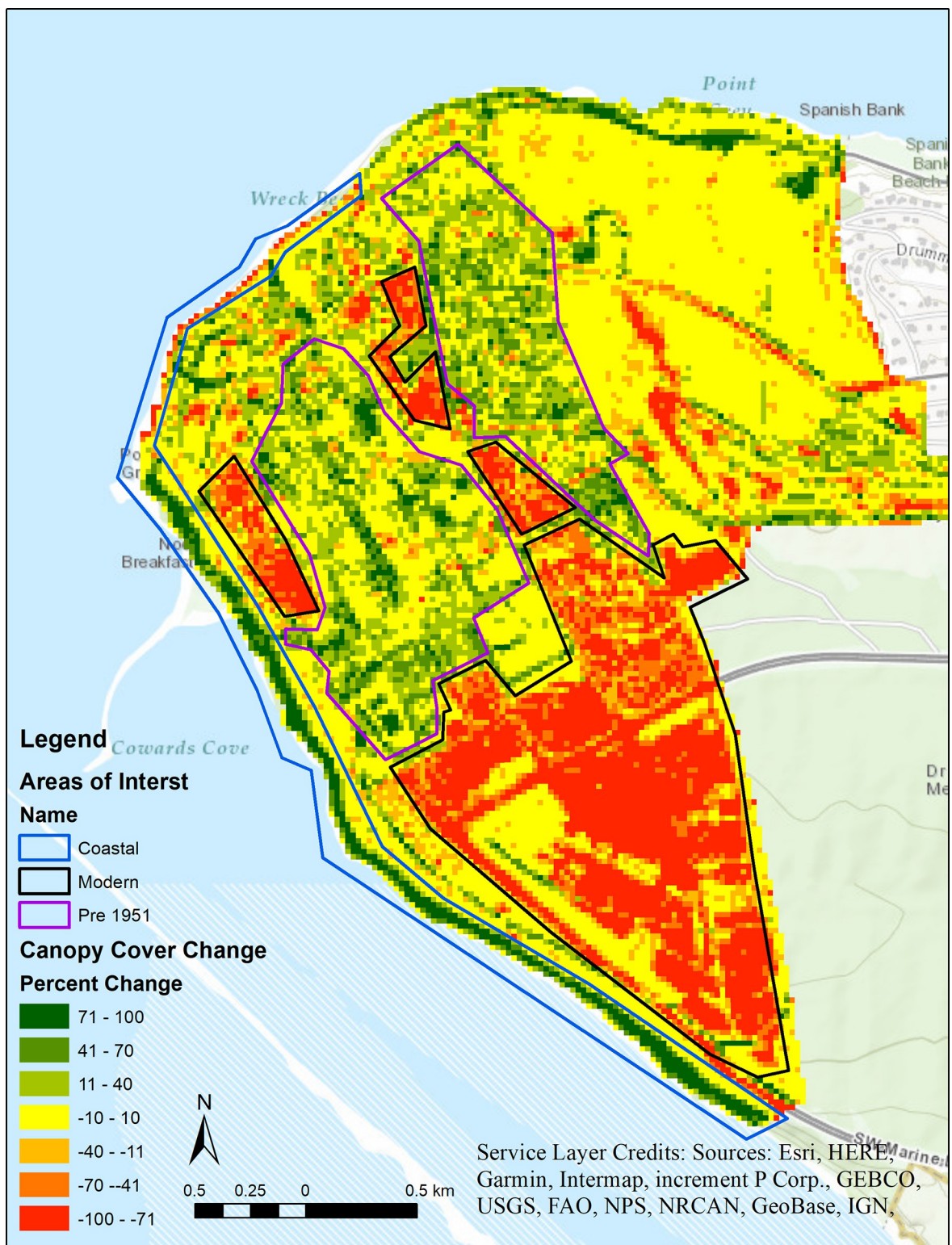

**Fig 4. Map showing the differences in forest cover at UBC's campus between 1949 and 2015.** Red indicates canopy reduction and green indicates an increase in canopy cover. The blue outline denotes the coastal area, black is the post 1949 (or modern) development and purple is the pre-1949 development area. Base map layer credit: Esri, HERE, Garmin, Intermap, increment P Corp., Gebco, USGS, FAO, NPS, NRCAN, GeoBase, ING (ArcGIS Licence 10.8.1).

of the data and the heavy tails on the Q-Q plots was expected. Both ends of the Q-Q plots were heavy indicating that the extreme canopy covers (<10% and >90%) were more common than mean values. This seems realistic for an urban setting where there will be patchiness in built-up areas and areas of high canopy cover, such as parks.

This study removed all features classified as trees that were less than 5 m. This height restriction was implemented to reduce the false classification of shrubs and is consistent with the FAO method of differentiating between trees and shrubs in classification processes [31], and was not validated further. However, it is likely that some small trees that have a height potential >5 m have been excluded in this research and some shrubs taller than 5 m have been included. The net impact of these inclusions and exclusions were deemed negligible due to the large data set size. An additional limitation in this research is the inability to ensure equality between the 1949 and 2015 aerial photos. The 2015 aerial photo cell size was adjusted to match the 1949 photos, but other attributes were not edited. Reduction of the cell size ensures that the overall photo resolution is the same but, cannot correct for differences in shadows, and color, leading to differences in TCC identification between the 2015 and 1949. These differences were therefor not accounted for in the validation process. Ideally, this research would have been able to validate the 1949 data directly with 1949 LiDAR data. Given this limitation, the two different years of aerial photos were made as equal as possible and future studies using these methods could provide further corroboration of the results from this research. Finally, this research found statistical equivalence within ±5.38% TCC. While statistically significant, this margin of error could have implications for practitioners. Urban forestry practitioners are often asked by municipalities to manage TCC within ranges of 1–2% and propose plans to ensure these goals are reached. From this perspective an error of ±5.38% TCC is significant. However, this research highlights an approach to monitoring long-term TCC trends that would otherwise be impossible with shorter data sets. To reduce error, we recommend that TCC change detection studies always be conducted with the best available data and only use two data types when other options are unavailable.

## Patterns of cover change

In the coastal areas, there was an increase in TCC on the western side of UBC and a decrease on the northern side. The reduction in TCC on the northern coastal area is likely due to cliff erosion. The erosion of the Point Grey cliffs has been a concern for UBC since the initial development of the campus in the late 1910's. Studies were conducted in 2002 and 2004 assessing the stability of the slopes [36] with a joint committee between UBC, the University Endowment Lands Administration, the British Columbia Ministry of Transportation and Infrastructure, and Metro Vancouver to conduct further studies in 2018 [37]. One of the purposes in collecting the 2015 LiDAR data of UBC was to accurately map the Point Grey cliffs to allow for comparisons of the slope with future data [38].

The western coastal area had an increase in TCC from 1949–2015. This coastal area has undergone changes that has also affected the tidal currents and patterns. In 1935 the North Arm Jetty, which now runs along the southern and western coast of UBC underwent major expansion. This jetty acts as a breakwater to calm the waters allowing for log booming [39]. The North Arm Jetty has undergone several additional changes including the addition of a wood debris processing facility in the mid-1960s [40]. Although the expansion of the jetty occurred 16 years before the time frame of this study, the effects of dampening currents would likely have long lasting impacts. Calmer waters reduce erosion and allow for the gathering of sedimentation and slow expansion of the coastline outwards. Over time, this could lead to an

increase in land available for tree growth causing an increase in TCC, matching the patterns seen at UBC.

In the areas developed pre-1949 there is an increase in TCC (Fig 4). The pre-1949 development is composed of educational buildings and residential single homes. Building types and development age are likely influencing the TCC gain [1,41,42]. Single-family homes have a positive correlation with TCC [41] and there is a lag between development and maximum canopy size [42]. This occurs because the urban forest canopy has time to recover from destructive development activities. This pattern is consistent with Berland's [1] findings that showed TCC increased with the age of urban development. Since 1949, areas of historical development have had time to recover.

The final area identified was the modern development area (Fig 4) in the southern portion of the study area. This area has undergone development within the last 15 years (initial plans adopted in 2005) [43]. Berland [1] found that TCC decreased with intensity of urbanization with conversion to urban land use causing the most substantial immediate loss of TCC. There is an immediate loss followed by a period of recovery. Since much of the development is recent, it is likely that the recovery period for TCC has not been realized. Future research could expand the study area to capture all of the Metro Vancouver regional district to determine if the patters detected hold consistent across a larger region.

## Conclusion

This research increases the ability to look at TCC changes when there are inconsistent data types available to researchers. This case-study was able to show that historical aerial photos can be used in conjunction with modern LiDAR data to examine TCC changes at small scales over times exceeding many remotely sensed data sources. Comparing between 2015 aerial phots and 2015 LiDAR data identified means of 54.94 and 54.70 respectively. Further statistical analysis identified no difference with each 0.05ha polygon having an equivalency of ±5.38% TCC. Concluding that the analysis of the two different data types lead to comparable results allowed for the analysis to be conducted between modern (2015) LiDAR data and historical (1949) aerial photos, and subsequent comparisons of the TCC. Future research at different locations with different data resolutions would be able to refine the data requirements for the methods uses. However, the data and methods used in this study indicate the ability to examine TCC in new ways through combining data types, expanding the ability to examine TCC over timeframes larger than is captured by a single data source.

## Acknowledgments

The authors would like to thank UBC geography for allowing access to their air photo collection, and UBC community and campus planning for access to the campus wide LiDAR data.

## Author Contributions

**Conceptualization:** Kathleen Coupland.

**Data curation:** Kathleen Coupland.

**Formal analysis:** Kathleen Coupland.

**Methodology:** Kathleen Coupland.

**Supervision:** Verena C. Griess.

**Validation:** Kathleen Coupland.

**Visualization:** Kathleen Coupland.

**Writing – original draft:** Kathleen Coupland, David Hamilton.

**Writing – review & editing:** Kathleen Coupland, David Hamilton, Verena C. Griess.

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
