## [Decision Letter · Decision Letter 0]

22 Mar 2022

PONE-D-22-02601Combining aerial photo and LiDAR data to detect canopy cover change in urban forestsPLOS ONE

Dear Dr. Coupland,

Thank you for submitting your manuscript to PLOS ONE. After careful consideration, we feel that it has merit but does not fully meet PLOS ONE’s publication criteria as it currently stands. Therefore, we invite you to submit a revised version of the manuscript that addresses the points raised during the review process.

We look forward to receiving your revised manuscript.

Kind regards,

Joe McFadden

Academic Editor

PLOS ONE

Journal Requirements:

4. We note that Figures 4 and 5 in your submission contain map images which may be copyrighted. All PLOS content is published under the Creative Commons Attribution License (CC BY 4.0), which means that the manuscript, images, and Supporting Information files will be freely available online, and any third party is permitted to access, download, copy, distribute, and use these materials in any way, even commercially, with proper attribution. For these reasons, we cannot publish previously copyrighted maps or satellite images created using proprietary data, such as Google software (Google Maps, Street View, and Earth). For more information, see our copyright guidelines: http://journals.plos.org/plosone/s/licenses-and-copyright.

a. You may seek permission from the original copyright holder of Figures 4 and 5 to publish the content specifically under the CC BY 4.0 license.  

5. We note that Figure 1 in your submission contain satellite images which may be copyrighted. All PLOS content is published under the Creative Commons Attribution License (CC BY 4.0), which means that the manuscript, images, and Supporting Information files will be freely available online, and any third party is permitted to access, download, copy, distribute, and use these materials in any way, even commercially, with proper attribution. For these reasons, we cannot publish previously copyrighted maps or satellite images created using proprietary data, such as Google software (Google Maps, Street View, and Earth). For more information, see our copyright guidelines: http://journals.plos.org/plosone/s/licenses-and-copyright.

Additional Editor Comments:

Two reviewers, both experts in this specific area of research, have provided detailed comments for significant revisions to the manuscript. A major concern, which also came out of my reading the the paper, was that it's unclear what point is being made about determining change over time with the modern LiDAR vs aerial photography data sets. Some of this may be a matter of clarifying the methods and intentions in quite a bit more detail. But other issues, such as those raised by Rev #2 about how you could ascertain the uncertainty of shrubs vs trees in the old photos, may require acknowledgements of the caveats of this type of analysis and discussion (based on literature) of how large a difference such errors might have been. All of the comments by Rev #1 and Rev #2 should be addressed in a revised manuscript, and I believe they will improve the quality and impact of the final article if they are fully considered.

Reviewers' comments:

Reviewer's Responses to Questions

**Comments to the Author**

1. Is the manuscript technically sound, and do the data support the conclusions?

Reviewer #1: Yes

Reviewer #2: Partly

2. Has the statistical analysis been performed appropriately and rigorously? 

Reviewer #1: Yes

Reviewer #2: No

3. Have the authors made all data underlying the findings in their manuscript fully available?

Reviewer #1: Yes

Reviewer #2: Yes

4. Is the manuscript presented in an intelligible fashion and written in standard English?

Reviewer #1: Yes

Reviewer #2: Yes

5. Review Comments to the Author

Reviewer #1: PONE-D-22-02601

Combining aerial photo and LiDAR data to detect canopy cover change in urban forests

In this study, the authors calculate and map tree canopy cover (%) change at the University of British Columbia (UBC) campus using airphotos and lidar data. Historical airphotos were from 1951 and lidar data were from 2015. They also use airphotos from 2015 to calibrate their results from the lidar. TCC was summed based on interpreted polygons of 0.05 ha to compare changes in the study area.

With additional revision, this could be a nice case study for tracking precise change through time in a particular region of a city. Description of statistics is mostly fine (see line by line comments). The scientific process and description are sound but I have a few concerns with its background and need more clarification on some of the results. However, in general the paper feels a overly brief and thin and could benefit from further analysis and discussion. There are also stray words and partial sentences in several places that require editing.

1. The authors used historical airphotos from 1951 and lidar data from 2015. However, the lidar data results were validated using 2015 airphotos exported from Google Earth. Why did the authors choose to use the lidar data to determine TCC change instead of using the 2015 airphotos directly? Is this forming the basis for related studies that would use both forms of data at different time points, and this is to show how airphotos and lidar can both be used? If the primary goal is determine TCC change, why use lidar if the existing methods work well with airphotos alone?

2. Need more description of the aerial photos from 1951 (lines 123-134). What was the approximate ground extent of the images (meters x meters)? How many photos were scanned? It would have been nice to be able to have a figure showing an example of the historical airphotos if possible (although I see that the data cannot be shared because of copyright permissions, unfortunately).

3. Figure 4 and Figure 5 have descriptive labels for colors for the amount of forest cover change (+High, + Med, etc.). What are the value ranges for these descriptive categories (+ x% TCC?). Descriptive ranges are not defined in text or caption, so it is hard to know what these categories actually mean. Also Figure 4 could be removed and just include Figure 5 since they are nearly the same figure. The lines for Coastal, Modern, and Pre 1951 areas could be made slightly thinner if the authors feel they are covering up too much of the change surface.

4. Much of the discussion is focused on the key areas identified in Figure 5. It would be beneficial to the results to quantify the TCC % change in each of these areas as a new figure or table in the results, and also the values for the whole study area. New figure could be a bar plot with mean TCC values for 1951 and mean TCC values for 2015 for each of the identified areas, with error bars for standard error, standard, deviation, or some quantile range. Otherwise it could be a new table instead that reports the mean values for each year and the % change. This would help provide more quantitative backing to the discussion “Patterns of cover change”. (Check out Figure 6 from Gillespie et al 2012 for a general idea: Gillespie, T.W., Pincetl, S., Brossard, S., Smith, J., Saatchi, S., Pataki, D., Saphores, J.-D., 2012. A time series of urban forestry in Los Angeles. Urban Ecosyst. 15, 233–246. https://doi.org/10.1007/s11252-011-0183-6)

5. Conclusion section feels incomplete, needs quantitative values from Results to support main takeaways. Conclusions should also include information regarding the TCC change results from the study.

Other general comments:

Line 43: Change to “TCC is”

Line 74: “short legacy” compared to…? This is fairly long record. Compared to development of over the 20th century? Longer term urban development? This needs a little more context. (It seems to be addressed somewhat in lines 95-99, but need something more here too.)

Lines 79-85: Spatial resolutions need to be listed in m, not m2. The finest scale MODIS pixels are 250 m on a side, not 250 m2. Similarly for RapidEye, spatial resolution is 5 m on a side, not 5 m2. For example, sentences can be rewritten as:

“Starting in 1999, MODIS produces images with spatial resolutions ranging from 250 m to 1 km. RapidEye is a group of 5 satellites launched in 2008 and has a spectral resolution a fine as 5 m.”

MODIS has coarser spatial resolution than Landsat; RapidEye has finer spatial resolution than Landsat. Should include references for MODIS and RapidEye.

Should include a sentence commenting on fine scale commercial satellite data (e.g., WorldView, Planet, etc), and why it was not used or readily accessible (likely cost).

Line 112: Please list a few of the dominant non-native planted tree species as well.

Line 136: Remove or correct “UBC,” at beginning of sentence

Line 139: Change “singletree” to “single tree”

Line 140: If the lidar data is rasterized, what was the spatial resolution of the rasterized canopy height layer from the lidar? Need a brief description of what is meant by “pre-processed”, is this the rasterizing step? If the lidar data is instead used to create tree crown object polygons here (ref Line 160), please state this clearly.

Line 148: Why was 0.05 ha selected? Based on tree cover mapping tool manual? Default or some other reason?

Line 169: should be “Shapiro-Wilk” test (revise elsewhere too, line 190)

Line 195: typo, “statistically”

Line 175 and Fig. 2: the values for TCC percent (54.94 in 2015 airphoto and 54.70 for 2015 lidar) are listed as medians in the main text (line 175) but are described as means in Figure 2 and in Figure 2 caption. Are they medians or means?

Line 191: check p-value, should be listed as p < (some small value), not p > (some value), to show significance.

Line 208: “For an average of” appears to be missing part of sentence

Lines 226-238: How do the resulting cover estimates for UBC compare to Vancouver as whole, if data is available from other urban forestry studies?

Line 285: Acknowledgments are missing are data sources, funding sources, etc.

Reviewer #2: The authors used historical air photos from 1951 and lidar data from 2015 to assess long-term changes in tree canopy cover. As someone who is interested in canopy cover change, I find this topic and approach interesting. However, I think there are noteworthy shortcomings in the analysis and the presentation of material that should be addressed before this manuscript is considered for publication.

At Line 204-210, the comparison of the 2015 air photos to the 2015 lidar is used to establish equivalency of canopy estimates derived from air photos vs. lidar. This, then, becomes the basis for claiming that the 2015 lidar data can be compared directly to the 1951 air photos. I think there is a false equivalency here, because the 2015 air photos are not the same product as the 1951 air photos. In my reading of the manuscript, this seems to be what the authors’ consider to be the primary contribution of the paper, but I’m not sure this is accomplished because there is no evidence that 2015 air photos are the same as 1951 air photos for the purposes of interpreting canopy cover.

At L142, the authors note that shrubs smaller than 5m were removed from the lidar classification of tree canopy. Can you similarly distinguish these shorter shrubs from trees in the 1951 air photos, and then exclude them from the canopy assessment? When interpreting the air photos, how can you be confident you are classifying a 6m tree as tree canopy and a 4m shrub as not tree canopy? A main message of the paper is that these two data sources (newer lidar and older air photos) can produce equivalent canopy data, but can you be sure about that when shorter vegetation was removed from the lidar dataset but the older air photos do not contain canopy height information?

Were multiple interpreters used to assess the interrater reliability of the canopy estimates? This is a common methodological practice, and is implemented nicely in recent papers such as Healy et al. (2022): https://doi.org/10.1007/s00267-022-01614-x

In the Introduction, I cannot tell what the authors mean by ‘urban forest.’ This term should be defined, as it is used in different ways in the literature.

L202: Is plus/minus 5.38% equivalent in practical terms? From a practitioner standpoint, what would it take to achieve a >5% gain in canopy cover? These numbers may be statistically comparable, but can you speak to the equivalency of these error ranges in practical terms?

There is a rich literature on tree canopy cover mapping and quantification, but many of the claims in the manuscript (particularly in the Intro) are not supported by appropriate references. Some examples:

-L62: “methods traditionally rely…” please provide citations

-L67: “numerous advances…” which advances? Provide references

-L70-71: provide examples of these contemporary assessments

-L72: “limiting factor…often is” - provide references to situations where this is the case

-L79: “commonly used” provide references

There are writing errors throughout that distract from the paper’s message. Some examples:

-In the title, should ‘aerial photo’ be ‘aerial photos’?

-In the data availability statement, no apostrophes are needed in ‘photos,’ and Google Earth should be capitalized.

-L49: afforestation has two f’s, and it is awkward to write “re- or a-forestation” – suggest writing out both terms

L108: this is not a complete sentence as written

L110: dominant, not dominate

L112: has seen, not see

L156: creating, not creates

L191: p > 0? This should be <, right?

L194: therefore

L195: statistically

L208: For an average of Since the 2015

…and so on

Beyond writing errors, there are many passages where the text could be reworded for clarity and conciseness. Examples:

-L30: delete ‘and increasing the understanding of’

-L265-275: This information could be written more succinctly

-L190: Do we need two paragraphs here essentially just stating that the distributions were not normal?

Line-item comments:

L23: Creating the polygons did not allow for TCC to be summed. Rather, there was an analytical approach for generating the canopy data. What approach was used? It is noted that the lidar and air photo data were ‘linked,’ but there is not a description of how they were linked. So when L27 says ‘the approach is suitable…’, I don’t know what the approach actually is.

L40: Average what? Annual decline? Average decline by state from 2002-2009? Please clarify

L42: If 22 of 45 were not statistically significant, then there was not statistical evidence to support the claim that those 22 were declining, right?

L72-85: It was already noted that air photos are preferred over satellite imagery for their high resolution and temporal depth. Is it necessary to dedicate a paragraph with detailed information about satellite platforms after already dismissing them in favor of air photos?

L82: It is not true that MODIS has finer resolution than Landsat

L175: Figure 2 shows mean values, but it is described in the text as median. Which is correct?

Fig 2 caption: What is CCr_valid? Also, the 54.94 and 54.70 values are given in the figure, in the figure caption, and in the text. There is no need for this redundancy.

I do not think Fig 4 is necessary, because the same information is repeated on Fig 5. Also, it would be more helpful to indicate quantities on the legend (not just high and low).

L260: Do you have evidence that the coastline expanded to the extent that would yield observable gains in canopy from 1951 to 2015?

6. PLOS authors have the option to publish the peer review history of their article (what does this mean?). If published, this will include your full peer review and any attached files.

Reviewer #1: No

Reviewer #2: No

---

## [Author Response · Author response to Decision Letter 0]

5 May 2022

Response to reviewer file has been uploaded. 

The excel document has three tabs. Tab 1 - Addresses general comments. Tab 2 - Addresses Reviewer 1 and Tab 3 - addresses reviewer 2.

---

## [Decision Letter · Decision Letter 1]

24 Jun 2022

PONE-D-22-02601R1Combining aerial photos and LiDAR data to detect canopy cover change in urban forestsPLOS ONE

Dear Dr. Coupland,

Thank you for submitting your manuscript to PLOS ONE. After careful consideration, we feel that it has merit but does not fully meet PLOS ONE’s publication criteria as it currently stands. Therefore, we invite you to submit a revised version of the manuscript that addresses the points raised during the review process.

We look forward to receiving your revised manuscript.

Kind regards,

Joe McFadden

Academic Editor

PLOS ONE

Journal Requirements:

Additional Editor Comments :

The revised manuscript was read by the same two experts who had reviewed the original submission. Both recommended additional revisions ("minor" level) before the paper can be accepted for publication. Some of the same important concerns were raised independently by both reviewers, and these must be addressed in the text of the paper. Notably, the reviewers also noted that some of their original concerns were not actually addressed by the authors in this revised version. They have brought those comments up again. My reading is that the concerns they have raised are accurate and reasonable. I would ask the authors to please respond to *all* of the second-round review comments *within the revised manuscript*, not only in the review response letter. For example, if there is a reviewer concern that cannot be fully resolved due to limitations of your data or methodology, that fact should be acknowledged within the body of the paper and you should set it in context as to how much that limitation would have affected your results. Items such as the percent TCC which were asked for by the reviewers the first time should be added (non-contiguous areas can be summed or averaged, or another approach that you can describe and justify).

Reviewers' comments:

Reviewer's Responses to Questions

**Comments to the Author**

1. If the authors have adequately addressed your comments raised in a previous round of review and you feel that this manuscript is now acceptable for publication, you may indicate that here to bypass the “Comments to the Author” section, enter your conflict of interest statement in the “Confidential to Editor” section, and submit your "Accept" recommendation.

Reviewer #1: (No Response)

Reviewer #2: (No Response)

2. Is the manuscript technically sound, and do the data support the conclusions?

Reviewer #1: Yes

Reviewer #2: Partly

3. Has the statistical analysis been performed appropriately and rigorously? 

Reviewer #1: Yes

Reviewer #2: Yes

4. Have the authors made all data underlying the findings in their manuscript fully available?

Reviewer #1: No

Reviewer #2: Yes

5. Is the manuscript presented in an intelligible fashion and written in standard English?

Reviewer #1: Yes

Reviewer #2: Yes

6. Review Comments to the Author

Reviewer #1: Combining aerial photos and LiDAR data to detect canopy cover change in urban forests

This study uses airphotos and LiDAR data to track changes in tree canopy cover at the University of British Columbia’s campus between 1951 and 2015, and the authors compare airphoto and LiDAR-derived methods to estimate tree canopy cover in 2015, specifically to evaluate how similar the tree canopy cover estimates are from the different data sources. The study suggests that LiDAR and airphoto methods can produce similar tree canopy cover estimates. The authors also evaluate the spatial patterns in tree canopy cover change between 1951 and 2015.

The manuscript is improved from the initial submission, and the authors have addressed most but not all the suggested comments. I believe the manuscript requires additional minor revision, and after these few comments are addressed, I believe it will be able to be accepted in PLOS ONE.

My 3 main comments are below, followed by a few line by line comments:

1. Line 214: I don’t understand the significance of the +/- 5.38% range for equivalency and agree with the other reviewer’s concern about it. The mean values from the 2015 air photos (54.942%) and the LiDAR (54.705%) are really very close, and I believe that their results are reasonably interchangeable based on what the authors have presented. However, this error range of over 5% for an “equivalency” isn’t very satisfying, and potentially allows for large differences in the imagery (theoretically a clearcut of up to 5% of the area between the 2015 airphoto and 2015 LiDAR acquisitions, if I’m interpreting this correctly). In this case, a test for equivalency should have a much tighter error range. I think this needs to be discussed more, especially because it’s highlighted again in the Conclusion. The key value that’s highlighted in the Abstract and elsewhere should perhaps be the actual difference in the estimated means, rather than this 5.38% range difference bound.

2. Line 293, Conclusion: “This research increases the ability to look at TCC changes when there are inconsistent data types available to researchers.” To me, this is what the authors intend to convey as the main contribution of the paper and should be placed at the beginning of the Conclusions paragraph. This framing should also be made more clear in the Abstract and Introduction. The results show that LiDAR-derived canopy maps and aerial photos can produce similar maps, and since LiDAR data is more accurate, new LiDAR can be used to compare with old airphotos using the technique presented here.

3. Conclusion section should list % tree canopy cover values in 1951 and in 2015 since this is a main takeaway of the study. Specify how these changes differ across the study area as discussed around Figure 4. Conclusions are still too general as written.

Line by line

Line 75: 50 years from Landsat is a long record, and the forest and forest management time scales comment is still too general. Should specify about time scales of historical development, historical airphotos, or the lifespan of trees or something similar

Line 86: MODIS is not a commercial satellite and does not have specific costs associated with accessing the data. RapidEye is operated by Planet but has free access through ESA accepted proposals. https://earth.esa.int/eogateway/missions/rapideye

Line 204-208: Suggest simplified rephrasing: “The non-parametric Wilcoxon signed rank test was used to test if the median canopy cover values from the 2015 aerial photos and the LiDAR data were different. We found p = 0.238, suggesting the medians were not significantly different.”

Line 209: Suggested rephrasing: “To further test for equivalency, we performed a TOST equivalence test…”

Line 236: grammar, “for the 64-year period”

Line 237: grammar, replace “it was” with “we”, easier to read with an active voice

Reviewer #2: I believe this comment from Reviewer 2 remains a key weakness of the manuscript. Downgrading the resolution of the 2015 images does not make total sense to me as a suitable fix for the issue: “the comparison of the 2015 air photos to the 2015 lidar is used to establish equivalency of canopy estimates derived from air photos vs. lidar. This, then, becomes the basis for claiming that the 2015 lidar data can be compared directly to the 1951 air photos. I think there is a false equivalency here, because the 2015 air photos are not the same product as the 1951 air photos. In my reading of the manuscript, this seems to be what the authors’ consider to be the primary contribution of the paper, but I’m not sure this is accomplished because there is no evidence that 2015 air photos are the same as 1951 air photos for the purposes of interpreting canopy cover.” I'm not sure what can be done to address this shortcoming at this stage, but I think it could be discussed in the Discussion section.

I appreciate the authors clarifying the classification of trees vs. shrubs in the text (in response to the second comment from Reviewer 2). But to me, I think that lack of validation of tree versus shrub classification is at odds with the stated goal of the authors “this study aimed to look specifically as combining different data sources and validating the results.” There was no validation of tree vs. shrub classification.

The authors were unclear on what was meant by the following comment, so allow me to clarify: “Is plus/minus 5.38% equivalent in practical terms? From a practitioner standpoint, what would it take to achieve a >5% gain in canopy cover? These numbers may be statistically comparable, but can you speak to the equivalency of these error ranges in practical terms?” I agree that 5.38% may not be statistically different. But is it different from the perspective of a practitioner? Many cities have set canopy goals. For example, the City of Vancouver put a lot of resources into meeting a goal to increase canopy from 18% to 22%, and they say that about 25,000 trees are needed to increase canopy by 1% (https://www.cbc.ca/news/canada/british-columbia/city-of-vancouver-to-meet-10-year-tree-planting-goal-1.5730251). So for Vancouver, it takes an ambitious, coordinated effort to increase canopy cover by 4%. At the same time, the authors are claiming here that plus/minus 5.38% is statistically equivalent. This is the difference between a statistical difference and a practical difference. In practice, for Vancouver a plus/minus 5.38% range represents the difference between exceeding a goal to increase canopy versus actually losing canopy. For an expected reader audience that is familiar with the difficulty in increasing tree canopy cover by even a few percentage points, I believe the practical significance of this degree of uncertainty should be acknowledged and discussed when interpreting the results.

No examples of contemporary assessments were added at L70: “This has precipitated contemporary remotely sensed TCC assessments to trend away from single spectrum aerial photo data towards these newer data types.” Please provide examples.

7. PLOS authors have the option to publish the peer review history of their article (what does this mean?). If published, this will include your full peer review and any attached files.

Reviewer #1: No

Reviewer #2: No

---

## [Author Response · Author response to Decision Letter 1]

22 Jul 2022

Thank you for the time spend reviewing and providing revisions to this research!

The Authors have included all suggested revisions in the manuscript and are outlined in detail in the "response to reviewers" excel document. 

The comments were insightful and the reviewers provided clarification regarding questions raised by the authors in the initial revisions.

---

## [Editor Report · Decision Letter 2]

10 Aug 2022

Combining aerial photos and LiDAR data to detect canopy cover change in urban forests

PONE-D-22-02601R2

Dear Dr. Coupland,

We’re pleased to inform you that your manuscript has been judged scientifically suitable for publication and will be formally accepted for publication once it meets all outstanding technical requirements.

Kind regards,

Joe McFadden

Academic Editor

PLOS ONE

Additional Editor Comments (optional):

Please proofread the final manuscript carefully and correct any spelling or grammatical errors.
---

## [Editor Report · Acceptance letter]

22 Aug 2022

PONE-D-22-02601R2 

Combining aerial photos and LiDAR data to detect canopy cover change in urban forests 

Dear Dr. Coupland:

I'm pleased to inform you that your manuscript has been deemed suitable for publication in PLOS ONE. Congratulations! Your manuscript is now with our production department. 

Kind regards, 

on behalf of

Prof. Joe McFadden 

Academic Editor

PLOS ONE